# Enhancing STEM Education by Integrating Research and Teaching in Photochemistry: An Undergraduate Chemistry Laboratory in Spectroscopy and Photochemistry

Eleanor J. Stelz-Sullivan [1], Jared M. Racca [1], Julia C. McCoy [1], Dana L. Charif [1], Lajmi Islam [1], Xiao-Dong Zhou [2], Barbara Marchetti [1,2,*] and Tolga N. V. Karsili [1,*]

1    Department of Chemistry, University of Louisiana at Lafayette, Lafayette, LA 70504, USA
2    Institute for Materials Research and Innovation, University of Louisiana at Lafayette, Lafayette, LA 70504, USA
*    Correspondence: barbara.marchetti1@louisiana.edu (B.M.); tolga.karsili@louisiana.edu (T.N.V.K.)

**Abstract:** Molecular spectroscopy and photochemistry constitute an integral field in modern chemistry. However, undergraduate level classes provide limited opportunities for hands-on experimentation of photochemistry and photophysics. For this reason, a simple laboratory experiment was designed that may be easily implemented into undergraduate teaching laboratories with the aim of introducing undergraduate students to UV/visible spectroscopy and photochemistry/photophysics and its possible applications. Samples of three unknown sunscreen formulations are given to students and they are asked to use a set of techniques to identify their molecular composition and to test their efficacy using basic laboratory equipment available to them. In particular, the students are asked to complete the following tasks: (i) sample preparation using solvent extraction to extract active ingredients from the sunscreen lotion, (ii) identify the extracted molecular sunscreen constituents by Thin Layer Chromatography (TLC) and UV/visible spectroscopy, and finally (iii) study their photostability by means of steady state irradiation coupled with UV/visible spectroscopy. The students were provided with the following tools for data collection: silica-backed TLC plates, a short-wave lamp (254 nm, for TLC analysis), a UV-Vis spectrophotometer with an associated computer and software, and an LED lamp (315 nm) to irradiate the samples. Combined TLC and UV-Vis spectroscopy allowed the students to identify the extracted ingredients. UV irradiation confirmed the photostability of sunscreens.

**Keywords:** photochemistry; thin-layer chromatography; undergraduate research; laboratory equipment/apparatus; spectroscopy

## 1. Introduction

Molecular photochemistry is a vital field in modern chemistry. Examples of important light-initiated reactions range from solar energy conversion to photosynthesis [1–5]. A particularly noteworthy example is in the field of heterogeneous and homogenous photocatalysis [5–10], in which light-irradiation initiates important chemical reactions that are otherwise infeasible under normal thermal conditions. Such reactions are essential for many important applications—e.g., the photoreduction of $CO_2$ into value-added fuels and chemicals [5,6,11].

Irradiation of a given molecular system with UV/visible (UV/vis) light may lead to electronic excitation. The nascent electronically excited molecule contains a greater total energy (cf. the ground state), is metastable, and thus decays in various ways. These are collectively described in a Jablonski diagram [12], i.e., a textbook illustration of the number of possible decay paths available to photoexcited molecules—in both the isolated gas phase and in bulk solution. Chemical filters are molecular constituents in sunscreen lotions that are designed to absorb incident UV radiation and dissipate the excess energy

as heat [13]. Several chemical filters have been synthesized to absorb and/or block harmful UV radiation [14]—the most common of which are displayed in Figure 1. Their geometric and electronic structures allow for rapid (on a sub-picosecond timescale) dissipation of the excess energy afforded by photoexcitation—to reform the parent structure with no detriment.

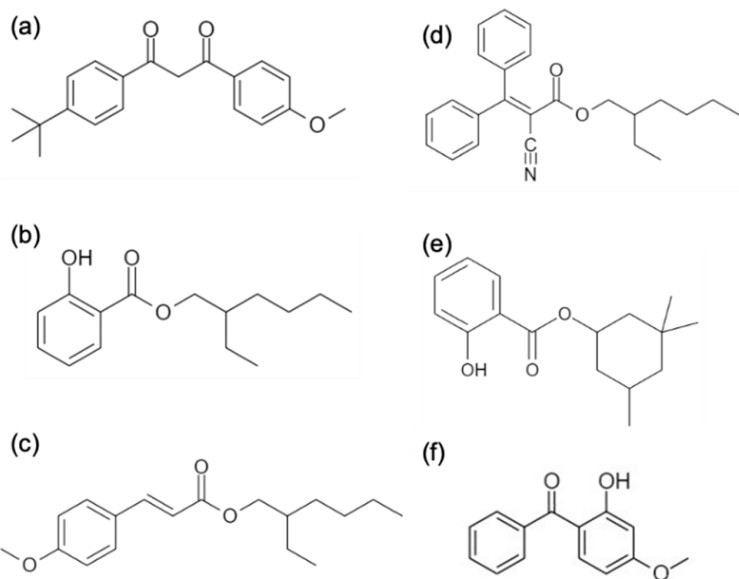

**Figure 1.** Common UV filters: (**a**) avobenzone, (**b**) octisalate, (**c**) octinoxate, (**d**) octocrylene, (**e**) homosalate, (**f**) oxybenzone.

Several studies have focused on the photophysics of molecular constituents in sunscreen formulations [13–16]. In each case, a sub-picosecond excited state decay and reformation of the parent ground-state structure is observed to dominate their excited state lifetimes—in a process that is commonly referred to as photostability. The excess energy is then dissipated as heat. Photostability is counter to the traditionally expected role of electronic excitation in synthetic/organic chemistry, in which photoexcitation is expected to promote a reaction. In the case of molecular sunscreens, the ultimate reformation of the ground state structure via photostability proceeds via internal conversion from an initially (or subsequently) prepared singlet excited state to the ground state [13,15–17]. Internal conversion is widely recognized to occur via conical intersections [18,19], which are regions of the molecular potential energy surface in which one or more electronic states of a common spin-multiplicity are degenerate. In some other molecular sunscreens, intersystem crossing has also been identified as a contributor to the ultimate reformation of ground state structure [20–24]. In these cases, intersystem crossing occurs with the greatest probability at degeneracy points between electronic states of distinct spin-symmetry, it is therefore a spin-forbidden process and thus occurs on a much longer timescale than internal conversion [12,25,26]. Specific nuclear motions drive a given molecular system towards such degeneracies between electronic states. In the example molecules given in Figure 1, the dominant nuclear motions that lead to such degeneracies include intramolecular proton-transfer as due to the presence of intramolecular hydrogen bonds (e.g., octisalate, homosalate and oxybenzone) and trans-cis isomerization (e.g., octinoxate and octocylene) [13,15,16].

The current work was inspired by a previous undergraduate experiment designed by Abney et al. [27], wherein the UV protective action of several sunscreen lotions and sunglasses is analysed using UV/vis spectroscopy. In this paper, we present a simple experiment which aims to develop valuable and transferable skills within the STEM (Science, Technology, Engineering, and Math) [28–30] field that students will be able to apply in other STEM classes and in industry at large. This experiment is aimed at encouraging students' self-efficacy in learning by enabling them to assimilate new understandings

of chemical processes into existing schema about how the world and industries around them operate. More specifically, the experiment exposes undergraduate students to basic separation techniques and spectroscopic methods to identify active ingredients present in selected commercial sunscreen samples. To the best of our knowledge, this experimental study represents the first reported undergraduate experiment in which solvent extraction, chromatography, spectroscopy and photophysics are combined in a single undergraduate teaching experiment, thus providing students with a complete procedure to analyse unknown samples which they prepared from an available commercial product. Specifically, the students will perform solvent extraction of chemical filters of three different sunscreen samples and will characterize them using Thin-Layer chromatography (TLC) and UV/vis spectroscopy. This experiment is designed to be implemented in lower division laboratory classes, such as general chemistry laboratories; however, with the addition of appropriate characterization techniques, it represents a good fit for implementation in analytical and instrumental laboratory courses. TLC and UV/vis spectroscopy were chosen as characterization techniques because they are relatively easy to implement in such laboratories and are routinely taught in general chemistry lecture and laboratory courses [31–42]. In particular, the UV spectroscopy (and photochemistry) of sunscreen has been previously explored at the undergraduate level in a handful of reported studies [27,43–45]. In this manuscript, we include the study of the photostability of the so-prepared of sunscreen samples: to perform such study, we use steady-state irradiation with UV light, coupled with broadband UV/vis probe [45].

## 2. Pedagogical Goals

### 2.1. Course Approach

Our current undergraduate chemistry curriculum at the University of Louisiana at Lafayette (UL Lafayette) caters for fundamental methodology, all of which align with the expectations of the standards set forth by the American Chemical Society. Although essential, photochemistry is largely limited to steady state electronic absorption spectra that complement synthetic efforts or to discuss optical properties of compounds, such as transition metal complexes. At its essence, photochemistry is chemistry that is initiated by light. Since most photo-induced chemical processes occur on fast timescales, methods for probing the chemistry in real-time are limited to sophisticated ultrafast laser technology. The experiment designed in this manuscript provides first-hand account of photochemistry in action, as well as training students in the separation and analytical techniques.

To successfully complete the experiment, students will be asked to complete a pre-laboratory assignment that covers topics associated with separation, solvent effects and photochemistry; the pre-laboratory assignment is completely at the discretion of the instructor, however, we suggest that a detailed pre-laboratory lecture should be delivered to students. During the pre-laboratory lecture, the instructor should emphasize the importance of photochemistry in topics that range from solar energy conversion to organic synthesis. Following completion of the laboratory session, students are expected to complete a laboratory report. Here, we present the results obtained during design and testing of the laboratory experiment.

### 2.2. Pedagogical Aims and Learning Outcomes

In a typical undergraduate chemistry curriculum, General Chemistry courses are taught across two semesters and include a combination of lectures and laboratory experiments. For instance, at UL Lafayette, prior to or alongside the General Chemistry Laboratory, students complete the General Chemistry II lecture course, in addition to the General Chemistry I lecture class, which is a prerequisite for the laboratory. The course- and module-level objectives for the General Chemistry Lectures and Laboratory are established in accordance with the expectations set by the American Chemical Society (ACS) and are set to provide students with a solid understanding of core concepts of chemistry. These courses are designed to expose student to fundamental understanding of matter classification and

composition, as well as atomic, molecular, and electronic structure, light-matter interaction, solubility and reactivity (e.g., acid-base reactions, ion exchange/precipitation reaction, etc.). Emphasis is then focused on the description and analysis of intra- and intermolecular interactions, vapor pressure, chemical equilibrium and kinetics, followed by thermodynamics and electrochemistry. Basic concepts concerning scientific method and experimental practices, such as experimental uncertainty, accuracy, precision, significant figures are also explored in these courses as well as some basic description of basic laboratory equipment, separation techniques (TLC and column chromatography), qualitative and quantitative analysis (titrations, pH sensors, UV/Vis spectroscopy, etc.). The adjoining laboratory course is design to sample topics and concepts explored in the lecture classes through experimental procedures. Solubility, thin-Layer Chromatography and spectrophotometric analysis of simple mixtures and solutions often feature in undergraduate chemistry laboratories so as to offer students the opportunity to appreciate chemistry concepts learnt in the lecture courses in an applied setting. Here, we present an experiment which combines three elements, namely solubilty/solvent extraction, TLC and spectroscopy/light matter interaction, in a single experimental procedure. The light-matter interaction is further extended with the addition of photo-irradiation studies in this procedure.

Thus, this experiment serves to provide students with hands-on experience with such topics. In this experiment, students will also explore how UV irradiation can initiate chemical or physical processes. The pedagogical aims of this experiment are to:

(1) Perform separation and analysis by chromatography, UV-Vis spectroscopy and photochemistry: students will be expected to identify the number of extracted active ingredients by performing TLC and their chemical identity by recording UV/Vis spectra of the samples.

(2) Perform photo-irradiation studies which will have the aim of testing the extent to which the extracted ingredient can resist UV exposure thus offer photoprotective action. To explore how a sunscreen component resists or degrades upon prolonged UV exposure, students will be asked to expose their sample to UV light for prolonged periods of time to test the efficacy of sunscreen. Student may be able to relate the sun protection factor (SPF) to the chemical composition of a sunscreen lotion.

## 3. Experimental Design and Overview

The experiment was designed and tested by a group of five undergraduate students, a graduate student, and faculty members who participated in conceptualizing the experiment, gathering the materials, writing the procedures, and data collection. The experiment was subsequently implemented in an undergraduate teaching laboratory for instructional purposes. The laboratory experiment is designed as a challenge for students to be undertaken in groups of three as there were three samples to be tested. Sample preparation and analysis was carried out using equipment which is usually available in undergraduate teaching laboratories such as materials for TLC (required), UV/Vis spectrophotometers (required), centrifuges (optional), vortex (optional), and GC-MS instrument (optional). These materials were chosen to make the experiment widely applicable, i.e., instructors at any institute would be able to implement the experiment without major adjustments. However, for testing the performance of the sunscreen samples in the last part of the experiment, a UV/Vis spectrophotometer needs to be coupled with a second light source for photoexcitation, such as LED lamp or short-wave lamp. Three different sunscreen lotions were selected for this experiment (see section below).

### 3.1. Choice of Sunscreen Lotions

Samples of each lotion were placed in a sample vial labelled sequentially from 1 to 3. In a teaching lab setting, no information regarding the composition, brand or SPF of the sunscreen lotions should be provided to the students prior to or at the time of the experiment to favor developing the students' critical thinking and problem-solving skills, as fundamental skills in STEM disciplines [29]. If the experiment falls outside of the students'

zone of proximal development (ZPD) [46] scaffolding could include providing them with some information concerning the composition or SPF of the sunscreen lotions. Each student in a team will be given a different sample (1–3) and asked to individually perform a series of operations and analyses, namely solvent extraction, TLC analysis, UV/vis spectroscopy and UV steady-state photoirradiation coupled with UV/Vis spectroscopy. Depending on the duration of the laboratory class, the experiment may be carried out across two different laboratory sessions; for instance, solvent extraction and characterization can be carried out on the first day, while the photophysics of the sunscreen lotions may be explored during the second laboratory session. The details of the experimental methods and concepts are described in the sections below, while the laboratory procedure is reported in the Supporting Information (Section S1).

### 3.2. Choice of Sunscreen Lotions

We selected three commercial sunscreens with a small number of active ingredients. Their SPF and composition are reported in Table 1.

**Table 1.** Brand, SPF (sun protection factor) and composition of samples 1–3.

| Samples / Information | Sample 1 * (Panama Jack) | Sample 2 ** (Coppertone Babies) | Sample 3 (Banana Boat) |
|---|---|---|---|
| SPF | 4 | 30 | 100 |
| # of UV filters | 1 | 2 | 3 |
| Avobenzone | | | ✓ (2.5%) |
| Octisalate (%) | | ✓ (5.0%) | |
| Octinoxate (%) | ✓ (2%) | ✓ (7.5%) | |
| Octocrylene (%) | | | ✓ (8.0%) |
| Oxybenzone (%) | | | ✓ (3.5%) |

* Contains coenzyme Q10 (antioxidant); ** contains ZnO (sun block ingredient).

One active ingredient (octinoxate) is common to samples 1 and 2. Sample 3 does not share any common active ingredients with the other two samples. Although any sunscreen can potentially be used, choosing a lotion with more than three UV filters can make it challenging to analyse the samples and interpret the result of TLC and UV spectroscopic analysis. Sample 1 contains coenzyme Q10 (CoQ 10), also known as ubiquinone, in addition to the UV filters and other inactive ingredients. The UV absorption spectrum of CoQ10 displays a prominent absorption band which is attributed to a $\pi* \leftarrow \pi$ electronic transition centred around the aromatic ring and which peaks in the UVC region (~275 nm in ethanol) [47,48]. Thus, CoQ10 does not act as an effective UV filter in the sunscreen lotions. The main purpose of CoQ10 is to enhance the antioxidant capacity of the skin, thus helping to prevent and/or reduce UV-induced damage and skin aging. Sample 2 contains zinc oxide as a sun block agent; the main role of this active ingredient is to block and/or scatter potentially harmful radiation, thus preventing UV-induced skin damage. Table 1 can be provided to the students with the answer keys of the experiment when the post-lab assignments are completed. It is to be noted that students are not expected to identify all extracted components in their samples; at the end of the analysis, they should however be able to identify at least one UV filter in sample 1 and 2, and two UV filters in sample 3.

### 3.3. Solvent Extraction

The sample were prepared according to a previously established procedure [27,49]. Briefly, ca. 200–300 mg of sunscreen lotions are weighted in two disposable 15 mL centrifuge tubes. A 2.00 mL aliquot of hot deionized water is then added to each test tube; a cloudy/milky suspension formed. The mixture is stirred for 2–5 min at high-speed using a vortex to obtain a more homogenous mixing of the components. Subsequently, a 10.00 mL

aliquot of isopropanol is added to the aqueous solution. The resulting solution now appears clear with the exception of some precipitate which is collected at the bottom of the tube. The mixture is vortexed at high speed for 2–5 min and subsequently sonicated for a further 10 min. The solutions are then centrifuged at high speed (3500 rpm) for 10–15 min. The supernatant fluid is used for further analysis. If the vortex, sonicator and/or centrifuge are not available, the mixtures can be manually shaken or stirred for a few minutes, but preparation of the samples must be carried out in advance and the suspension left to settle for at least a day to facilitate separation of any undissolved material from the solution [27]. This is particularly important for the UV analysis so as to minimize light scattering from undissolved materials which may arise if the precipitate is still suspended in solution.

*3.4. Chromatographic Analysis*

TLC analysis was performed on the supernatant fluid to (i) ascertain whether the solvent extraction has been carried out successfully, and (ii) discover the number of substances extracted during the sample preparation phase. The retention factors ($R_f$) of extracted substances were also calculated as an exercise for students with the purpose of training and education. For this purpose, a mobile phase of 15 parts of hexane to 2 parts of acetone was prepared and placed in a 100 mL beaker. The beaker was covered with plastic film and set aside to serve as the developing chamber for the TLC analysis. Standard silica gel 60 TLC plates (on aluminium substrate) were used as the stationary phase. The TLC analysis was then performed on samples 1–3. Preparation of the TLC plated needs to be carried out with care; it is important to instruct the students that the supernatant solution might contain sunscreen-extracted ingredients in small concentrations. Therefore, in this study, a shortwave lamp (254 nm) was used to check the TLC plates *prior* to developing them to ensure that enough sample had been deposited. If otherwise, more sample solution was applied to the plate to ensure that development of the plate returned reliable results. TLC was used only to perform a qualitative analysis, i.e., to ascertain that some ingredients have been extracted during sample preparation and approximately identify the number of such components. TLC could in principle be used to perform a *tentative* assignment, if the $R_f$ values of appropriate standard compounds are known. However, we recognized that this technique is not the most reliable for this purpose, therefore, it was not employed for assignments of extracted ingredients. Nonetheless, students may be asked to calculate the $R_f$ of any spot identified through TLC analysis for instructional purposes, as this would provide a more solid understanding of substance separation in mixture, basic working mechanisms of chromatography and its analysis. To this end, the distances travelled by the mobile phase ($d_m$) and by the compounds ($d_s$) were measured, and the $R_f$ for each compound are calculated as the ratio between the two values.

*3.5. UV Spectroscopy*

A Vernier UV-VIS Spectrophotometer was used to acquire UV absorption spectra of the samples in water/isopropanol and reference compounds in isopropanol. The samples (supernatant fluid) were diluted until the maximum absorbance was less than 1. The UV absorption spectrum of each diluted solution was recorded between 220–800 nm. In a teaching laboratory setting, we suggest providing students with the UV-Vis absorption spectra of avobenzone, octinoxate, octisalate, octocrylene and oxybenzone for comparison with those of samples 1–3. In this study, high purity avobenzone, octinoxate, octisalate, octocrylene and oxybenzone were dissolved in isopropanol, and their UV absorption spectra were then recorded. The absorption spectra of the samples were compared to those of the reference compounds. Two different approaches can be used for identification of UV filters by comparison with the standards. Method 1, e.g., the simplest approach, requires students to simply compare the maxima of the absorption spectra of the reference compound (or their profile) to those of the sunscreen samples. This aids the assignment of the absorption features of samples 1–3 to specific UV filters; with this method, the present of at least one UV filter will be ascertained in samples 1 and 2, while at least two UV

filters should be discernible in sample 2. Method 2 requires a more complex approach that consists of a basic fitting procedure to be performed after Method 1. Such fitting procedure can be carried out in an Excel spreadsheet to aid the assignment and was performed as it follows. A first comparison between the absorption spectra of the samples and reference compounds (as in Method 1) was first carried out to obtain an initial guess for the active ingredients present in sample 1–3. From this, the UV absorption spectra of the relevant references were summed to obtain a cumulative absorption spectrum. This cumulative absorption spectrum was plotted as overlaid and compared to that of sunscreen sample of interest. If the agreement was not satisfactory, the overall absorption spectrum was modified by gradually adjusting the relative weights of the individual references until satisfactory agreement was obtained. The fitting procedure also includes an estimation of the coefficient of determination ($R^2$) and the Root Mean Square Error (RMSE) as indicative the goodness of the fit; if students are not familiar with the concept of $R^2$ and RMSE, the instructor could provide a basic explanation of their meaning. In general, we recommend that the fitting procedure is implemented in an introductory class if students are adept with the use of Excel. If students are not familiar with the use of this analysis and graphing program, as often true in introductory classes, then a qualitative comparison between the spectra of the samples and those of the reference compounds may suffice for assignment.

### 3.6. Photophysics of Sunscreen Lotions

Finally, steady-state irradiation was carried out to illustrate the photostability of commercial sunscreen formulations. The set-up used in this experiment was described in our previous publication [45]. Briefly, it consists of a Vernier UV-Vis Spectrophotometer coupled to a high power LED lamp irradiating the sample cuvette. A 1.00 cm quartz cuvette with 5 clear windows (the 4 sides and the bottom) was used; irradiation occurred through the optically polished bottom of the cuvette. The set-up was fully enclosed in safety shielding ('UV chamber') to prevent exposure to UV light. A schematic representation of the set-up is illustrated in Figure 2.

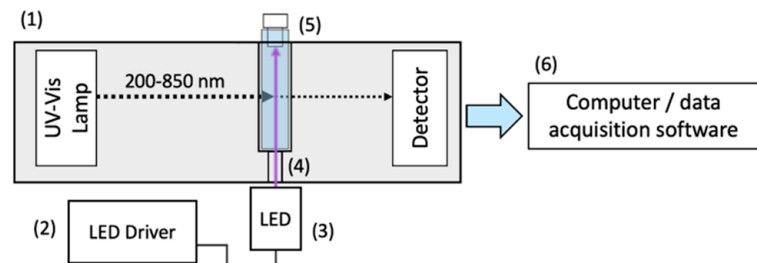

**Figure 2.** Schematics (side-view) of the experimental set-up: (1) UV–VIS Vernier Spectrophotometer, (2) LED Driver, (3) LED lamp, (4) optical output of LED (315 nm in this study), (5) 10.0 mm fluorometric cuvette (quartz, bottom polished), (6) computer for data acquisition.

The LED provides steady-state irradiation at selected wavelengths while the spectrophotometer is set to acquire a UV/vis absorption spectrum of the sample every 120 s. In this study, the samples were irradiated at 315.0 nm (Thorlabs LED315W, FWHM = 11.0 nm, 1.0 mW optical output) for 60 min while monitoring the UV/vis absorption spectra as a function of the irradiation time. The irradiation wavelength was chosen primarily because it falls within the UV-Vis spectra of the UV filters, while most other organic species which may be present in the extracted samples are likely to absorb light of shorter wavelengths (<300 nm). Dependence on the UV excitation wavelength was not explored in this study. Our photoirradiation assembly has a cost of ~$2800 (spectrophotometer included); we recognize that, if budget for implementing the experiment is limited, it may not be possible to purchase several set-ups to allow students to work individually on the apparatus, especially if spectrophotometers are not already in use the class. Possible cost-effective ways to

circumvent this issue may be to allow students to share the available set-ups by working in groups or undertake the experiment in rotation during the semester.

## 4. Hazards

The students are expected to be aware of good health and safety practices as they pertain to a chemical laboratory setting. At UL Lafayette, our first laboratory session involves a safety lecture. Safety spectacles and a laboratory coat must be worn during the laboratory experiment, as well as closed-toes shoes and other laboratory-suitable attire. When using volatile solvents use of fume hood is required.

## 5. Results and Discussion

### 5.1. Chromatographic Analysis

The first step of the experiment is to develop TLC plates for samples 1–3 and to reveal the spots under a UV shortwave lamp. The $d_m$, $d_s$ and calculated $R_f$ obtained by TLC are given in Section S2 of the Supporting Information. The interpretation of the results is briefly described here. In this work, sample 1 appears particularly dilute; therefore, extra care was needed in preparing the TLC plated for development. The sample displayed at least one spot under the UV light ($R_f = 0.33 \pm 0.09$); this spot appears streaking in some plates, while in others a second spot appears fainter at lower $R_f$. It is therefore concluded that at least one, possibly two substances are present in extracted sample 1. Sample 2 presents at least two spots appearing at $R_f = 0.036 \pm 0.03$ and $0.44 \pm 0.02$. At least two additional faint spots seem to be present at higher and lower $R_f$ value. Evaporating some of the solvent, e.g., by gently heating the sample 1 and 2 for a few minutes, could allow visualizing of the less evident spots. For instance, in sample 2, this allowed visualization of a more evident spot at $R_f = 0.63 \pm 0.03$. Finally, sample 3 showed three distinct spots ($R_f = 0.50 \pm 0.03$, $0.58 \pm 0.03$ and $0.78 \pm 0.03$), suggesting that at least three different compounds are present in solution. We recognize that detection and definitive identification of UV filters or any other compounds (including CoQ10 and ZnO, as present in a subset of sunscreens) possibly present in the samples requires more sophisticated analytical techniques. If the experiment is implemented in higher division classes, such as analytical or physical chemistry, additional techniques (e.g., column chromatography, HPLC, GS/MS and H-NMR) can be used to determine the composition of the samples and/or separate the different compounds for further analysis. However, since such techniques are typically not used in introductory chemistry classes (which this experiment is aimed at), we consider TLC as an appropriate technique to provide at least an indication of successful extraction of active ingredients from the original, commercial sunscreen lotions. Additionally, in depth analysis of samples with the use of one additional (high level) analytical techniques is however viable and permitted within a 3–4 h laboratory period, if the experiment is carried out across two different laboratory sessions.

### 5.2. UV/Vis Spectroscopy

Following the TLC analysis, the students will be asked to confirm their first assignment using UV spectroscopy. Unlike IR spectroscopy, UV spectroscopy is generally not the most reliable tool for identification of substances as it does not return specific structural information. However, in the current case, it represents an easy diagnostic technique since each UV filter displays a distinct electronic spectrum in the near UV. This can be used as a spectroscopic signature for the identification of the compounds in our samples. Figure 3 presents the absorption spectra of the three samples.

Firstly, the UV/vis spectra of the three samples were recorded. These are displayed in Figure 3 of the main text. The first analysis reveals that the spectra of samples 1 and 2 (blue and cyan solid lines) are very similar. Both spectra display an intense absorption band peaking at ~310 nm. On the short wavelength edge of this band, the absorption spectrum for sample 1 is slightly broader cf. sample 2. A second absorption band peaking at shorter wavelengths (~225 nm) is observed for both samples.

The UV absorption spectrum for sample 3 appears substantially different when compared to the others. The absorption spectrum is much broader and displays at least two distinct but overlapped absorption bands in the near UV/vis portion of the spectrum. These two main absorption bands of this sample peak at ~323 and ~290 nm. Additional intense bands are observed at shorter wavelengths. Under the photoprotection perspective, sample 3 may be considered a broader spectrum UV filter due to the broader range of wavelengths absorbed by its active ingredients which extend all the way to the visible portion of the electromagnetic spectrum.

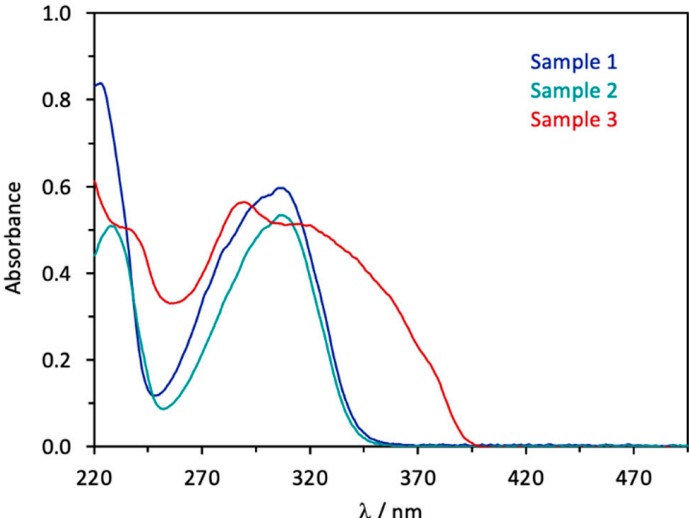

**Figure 3.** UV–Vis spectra of the three samples.

We will now turn our discussion to the identification of the UV filters by comparison of our results with the spectra of the reference compounds. It is to be noted that UV analysis needs to be carried out with due attention. When compared to TLC, UV spectroscopy offers a more sensitive tool for detecting small amounts of substances which may be present in the sample solutions. It is therefore anticipated that chromophores other than the UV filters may contribute to the overall spectra of the three samples. Since the supernatant fluid was analysed without further separation or purification, some inactive ingredients in the original sunscreen lotion are likely to be dissolved in the analysed samples, as also suggested by TLC analysis. This is likely to occur since inactive ingredients in the lotions may also be partially soluble in isopropanol and thus extracted during the sample preparation phase. However, among various ingredients in the sunscreen lotions, the UV filters are the only substances which are likely to absorb in the UVA region. For these reasons, we recommend disregarding the portion of the UV spectra at wavelengths shorter than 270 or 250 nm, since traces of unwanted compounds are more likely to contribute to the absorbance in this region of the electromagnetic spectrum,

The students will then be provided with (or asked to record) the individual UV-Vis absorption spectra of avobenzone, octisalate, octinoxate, octocrylene and oxybenzone (reference compounds) in isopropanol. These absorption spectra are shown in Figure 4a–c together with the those of samples 1–3. By qualitative comparison (Method 1), it is evident that the peak wavelength and shape of the absorption spectra of sample 1 and 2 are in good agreement with that of the reference octinoxate. Sample 3 presents a distinct profile with a prominent band at ~290 nm which is a characteristic feature of the UV filter oxybenzone. On the other hand, the absorption spectrum of this sample extends to ~400 nm. The only UV filter which absorbs light at this region is avobenzone. Figure 4 also shows how the UV absorption spectra of the reference compounds can be summed to reproduce those of samples 1–3 (method 2). This figure shows that the absorption spectrum of octinoxate (blue dashed line) overlaps well with that of sample 1. The sample however displays a broader profile in the short wavelength side; here we added a small

contribution arising from the absorption of CoQ10 (absorption spectrum reproduced from references [47,48]) to better reproduce the experimental results for sample 1. Although the absorption spectrum of sample 2 overlaps to a satisfactory extent with that of octinoxate, an additional contribution arising from octisalate (green dashed line) returns a better agreement. The absorption spectrum of sample 3 includes contributions of oxybenzone (red dashed line) and avobenzone (yellow dashed line), as established above, however addition of a third UV filter (octocrylene, dark red dashed line) returns an excellent agreement.

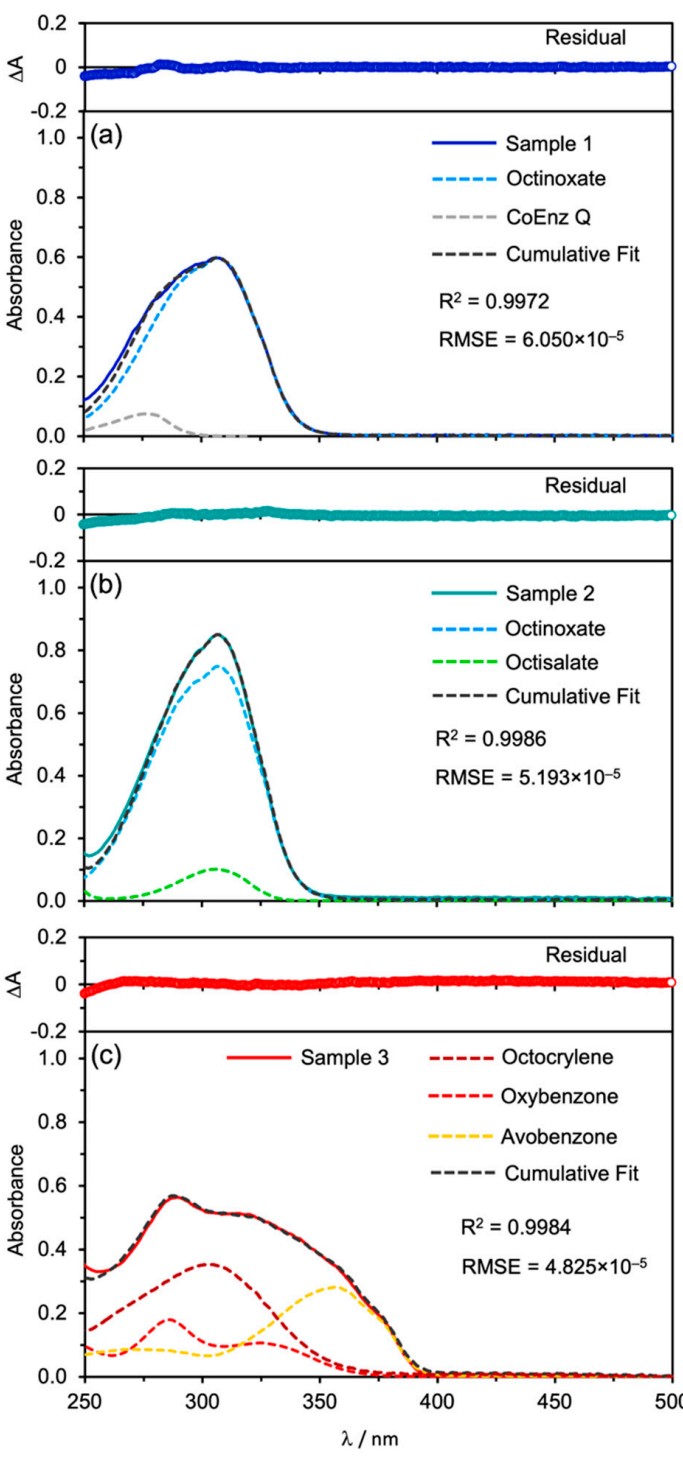

**Figure 4.** Fittings of UV–VIS spectra of the three sample, (**a**) sample 1, (**b**) sample 2, (**c**) sample 3.

### 5.3. Steady-State Irradiation and Photostability

The final part of the laboratory exercise consists of exploring the photostability of sunscreens using a simple UV steady-state photoirradiation experiment. A previous undergraduate experiment designed by Abney et al. [27] explored the photoprotective role of UV filters and their efficacy by considering the range of UV-vis wavelengths absorbed (or scattered) by a variety of commercial sunscreen lotions and sunglasses lenses. The authors concluded that broader spectrum sunscreens should be considered more effective in protecting skin (or eyes) from harmful UV solar radiation. Although this is true to some extent, an important quality of sunscreens is their ability to withstand prolonged UV irradiation without decomposing to form harmful chemical species. As such, both a broad UV absorption spectrum and photostability are essential to determine the efficacy of sunscreens. Here, we used our in-house UV photoirradiation set-up (see Figure 2) to assess the photostability of the sunscreen samples. The resulting UV-Vis absorption spectra recorded as a function of irradiation time are displayed in Figure 5. Various research studies have focused on determining the mechanisms of photostability of individual UV filters in solutions by means of ultrafast absorption spectroscopy [13,15,16]. These studies concluded that internal conversion from a photoexcited state to the ground state facilitates recovery of the original molecule. Some evidence of incomplete recovery of the initial ground state molecules has been observed in ultrafast pump-probe absorption studies, especially under UV-B irradiation, for instance in the case of oxybenzone [50]. This incomplete recovery of the initial molecule is attributed to the formation of a small quantity of photoproducts, suggesting that some degradation occurs upon prolonged irradiation of the UV filter.

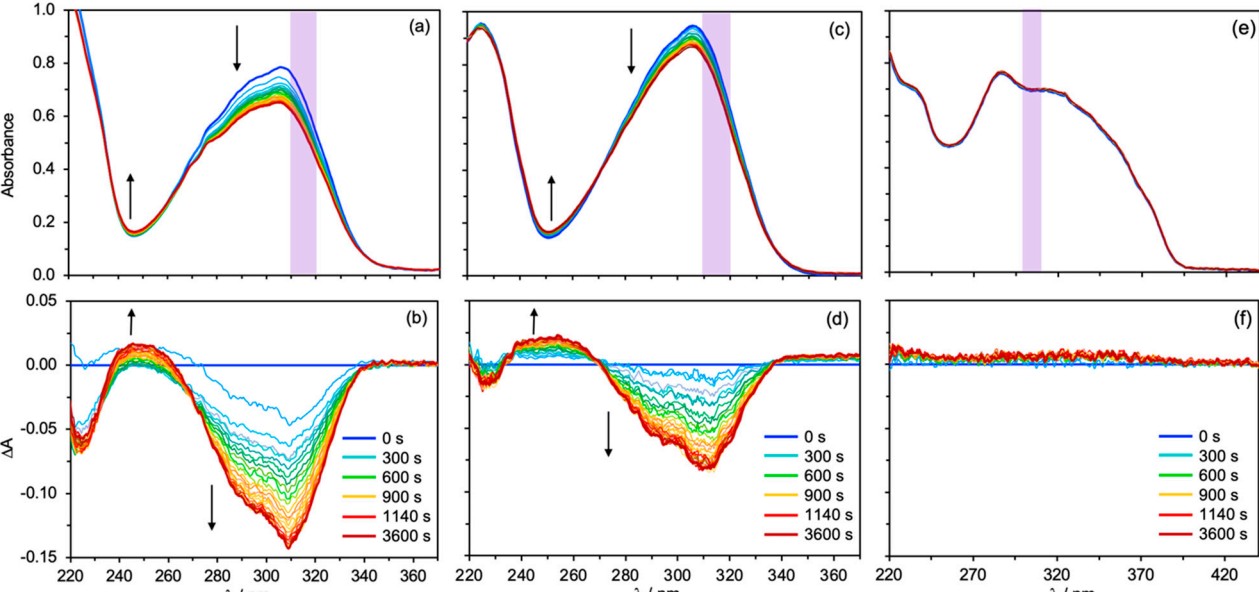

**Figure 5.** UV–Vis absorption spectra as a function of irradiation time of (**a**) samples 1, (**c**) sample 2 and (**e**) sample 3 at $\lambda_{irr}$ = 315 nm. The irradiation window is highlighted in purple. Differential UV/Vis absorption spectra of (**b**) sample 1, (**d**) sample 2 and (**f**) sample 3.

Femtosecond laser experiments as those cited above are not a viable option for studying photochemistry in an undergraduate laboratory. However, alternative experiments have been proposed and can be used for this purpose [51–54]. In our previous publication in *J. Chem. Ed.* [45], we demonstrated that our pump-probe experiment is able to promote photochemical reactions in some organic compounds like phenol, while a more photostable compound, i.e., the UV filter oxybenzone, was able to withstand UV irradiation without photodegradation. We can therefore conclude that our setup is appropriate to effectively test the photostability of organic compounds including UV filters that are included in sunscreen formulations.

In this study, the students focused on exploring the photostability of a mixture of UV filters extracted directly from the lotions, rather than that of the individual (high purity) active ingredients purchased by a commercial supplier. Thus, the aim was to explore the synergistic action of the combination of UV filters in absorbing UV light and dissipating the excess energies without any detriment to its molecular structure. To this end, steady-state irradiation of samples 1–3 was performed using a continuous LED lamp emitting UV light at 315 nm, as stated in the methodology section. Irradiation was performed for a total of 60 min which we deemed to be an appropriate duration for a teaching lab section. Depending on the time allocated for the experiment, the irradiation time can be considerable shorter; this could be advisable if only a limited number of set-ups are available, and students are therefore required to share the instrument. Figure 5a,c of the main text shows the time-resolved spectra of samples 1 and 2 under steady-state irradiation, while Figure 5b–d reports the equivalent results in the form of differential spectra, obtained according to Equation (1):

$$\Delta A(\lambda) = A_t(\lambda) - A_0(\lambda) \tag{1}$$

where $A_0(\lambda)$ is the absorbance at the wavelength $\lambda$ and before the irradiation begins (time $t = 0$), while $A_t(\lambda)$ absorbance at the same wavelength $\lambda$ and at a specific irradiation time $t$.

As observed in this figure, both samples 1 and 2 display minor changes in the UV absorption as function of irradiation time $t$, with the absorbance decreasing at $\lambda_{max}$ and increasing ca. 250 nm, as illustrated by the arrows. At least one isosbestic point can be seen at ~270 nm implying the presence of a second species which is being formed in solution upon UV irradiation. In accord with previous studies, the spectral evolution can be attributed to photo-induced isomerization by which the *trans*-octinoxate isomerizes to the *cis* conformer [55,56]. Although motion along the isomerization coordinate is responsible for promoting internal conversion to the ground state to recover the initial *trans*-octinoxate, experimental results suggest that a minor fraction of *cis*-octinoxate is also formed. As observed in these spectra, *trans* and *cis*-octinoxate do not undergo any further photoreaction within 60 min of irradiation and under the current conditions, implying that both conformers are photostable and, therefore, act as good sunscreen agents. However, although no radicals or highly reactive species are formed upon irradiation, *cis*-octinoxate has a lower molar absorptivity coefficient and its formation leads to a decrease in the absorbance of the sample. This implies that isomerization decreases the photoprotective performance of sunscreen samples 1 and 2—but only by a small extent. From the spectra, it can be inferred that photoirradiation of sample 2 results in a lower yield of *cis*-octinoxate conformers, possibly due to the synergistic action of the second UV filter (octisalate) present in the sample. Photokinetic curves were not extrapolated from the spectra in Figure 5a,c, but for higher level chemistry courses, such as physical chemistry, the absorbance at $\lambda_{max}$ and 250 nm can be plotted as a function of irradiation time and the resulting curves could be fit to kinetic models to obtain the photoisomerization rate coefficients.

Photoirradiation spectra of sample 3 are reported in Figure 5e,f of the main text. In this case, no detectable changes in the absorption spectrum are observed upon sustained UV irradiation. This is indicative of the remarkable photostability of the sunscreen formulation—e.g., its capability to withstand prolonged UV excitation, which confirms that the combination of UV filters and their synergistic action make it an extremely effective UV protective sunscreen lotion.

At the end of this laboratory experiment, students should recognize that, although all samples are able to withstand irradiation over a 60-min time window, sample 3 can be regarded as the most effective for the following reasons:

1.  Its absorption spectrum provides UV protection over a broader UV range
2.  No photochemical reaction is observed
3.  Its absorbance is not altered by any photophysical process (e.g., photoisomerization)

Briefly, no drastic changes are observed in the UV-Vis absorption spectra of samples upon UV irradiation. However, it is clear that sample 3 is the most photostable, as expected based its higher SPF.

We recognize that analysis of post-irradiated sample with the aim of detecting photoproducts by analytical methods could represent an interesting expansion of this experiment if implemented in higher level classes. It is to be considered however, that due to the use of very diluted solutions (in the micromolar regime), identifying possible photoproducts by analytical techniques (including GC-MS and/or NMR) maybe be extremely challenging, unless large volumes of sample solutions are irradiated. The low concentration of reactants, coupled with small volumes of solutions typically irradiated (~1–3 mL) in this (and analogous experiments) results in extremely small yields of photoproducts even in cases in which photoproducts are produced efficiently; this precludes straightforward identification of such products formed upon photoirradiation.

## 6. Conclusions

In this study, we constructed a simple experiment to be implemented in a general chemistry laboratory. The experiment was designed to include a series of techniques which students need to perform to solve a scientific challenge. The laboratory experiment is built to incorporate solvent extraction, chromatography, and spectroscopic analysis of selected sunscreen samples. The last part of the experiment represents a simple yet effective application of a UV steady-state photoirradiation method coupled to UV-Vis spectroscopy; here, wavelength selective photoexcitation of a sample is followed by broadband UV probe, to study any eventual chemistry occurring upon excitation. The sunscreen samples studied in this experiment showed remarkable photostability. Two of the analysed samples presented some minor changes in their UV absorption spectra. Although identification of photoproducts responsible for such spectral changes by analytical methods is not straightforward, especially in undergraduate teaching settings, an additional computation chemistry component could be included in this experiment for this purpose if time permits. Similarly to our previous study [45], students may be asked to consider possible products being formed upon photo-excitation. Afterwards, students could be asked to then use Density Functional Theory calculations to optimise the ground state of the parent molecule and possible photoproducts, followed by computation of vertical excitation energies and oscillator strengths. Comparison between the excitation energies and oscillator strengths with the experimental spectra could provide students with insights into photophysics and photochemistry of the explored systems, while analysis off molecular orbitals could be used to illustrate orbital character of electronic promotions and how they relate to the value of oscillator strength and intensity of absorption features.

**Supplementary Materials:** The following supporting information can be downloaded at: https://www.mdpi.com/article/10.3390/educsci12100729/s1, Section S1: Laboratory Manual; Section S2: The TLC results.

**Author Contributions:** Conceptualization, B.M. and T.N.V.K.; Data curation, E.J.S.-S., J.M.R., J.C.M., D.L.C., L.I. and B.M.; Formal analysis, E.J.S.-S., J.M.R., B.M. and T.N.V.K.; Funding acquisition, X.-D.Z., B.M. and T.N.V.K.; Methodology, B.M. and T.N.V.K.; Resources, T.N.V.K.; Supervision, B.M. and T.N.V.K.; Validation, E.J.S.-S., J.M.R., D.L.C., L.I., X.-D.Z., B.M. and T.N.V.K.; Writing—original draft, E.J.S.-S., B.M. and T.N.V.K.; Writing—review and editing, J.M.R., X.-D.Z., B.M. and T.N.V.K. All authors have read and agreed to the published version of the manuscript.

**Funding:** The work is supported by the National Science Foundation, under grant numbers 2003422 and NSF-2119688.

**Institutional Review Board Statement:** Not applicable.

**Informed Consent Statement:** Not applicable.

**Data Availability Statement:** The data underpinning this article will be made available upon reasonable request from the corresponding author.

**Conflicts of Interest:** The authors declare no conflct of interest.

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
