# Peer review of "Enhancing STEM Education by Integrating Research and Teaching in Photochemistry: An Undergraduate Chemistry Laboratory in Spectroscopy and Photochemistry"

_education, doi:10.3390/educsci12100729_

Round 1
Reviewer 1 Report
The research seems up-to-date and interesting in terms of its subject. Although the research title is not too long, it is suitable for the purpose. It was also bad that the abstract of this research was not in the structured abstract form. In other words, it would not be good not to write a summary including the purpose, method, data collection tool and results in the summary. However, the research findings should be briefly mentioned in the abstract and the term "we" should not be used. Third language is used in research articles. The introduction of the research is appropriate in terms of literature, but it is not sufficient. The introductory part of the research is also not sufficient in terms of subject area. The research is related to STEM, but the concept of STEM has not been adequately addressed. For example, the study "Tezer, M. (2019. The Role of Mathematical Modeling in STEM Integration and Education. In Theorizing STEM education in the 21st century. IntechOpen") in the field of STEM and its theory and similar research could be included. The bibliographies used are up-to-date. Therefore, the use of new bibliography in the introduction and discussion sections of the research has enriched the research. Purpose and sub-objectives were written in line with the findings. The research method is not well written. Experimental research method was used in the research. But in the research, it would be better to write the experimental research design for this experimental study. The research could be used in STEM education, but it could also have an educational aspect. It would be better if the research included students' opinions and the validity-reliability of the data collection tool was also included. In other words, the educational dimension of the research is lacking. My recommendation is to add students' views to the research through qualitative methods. Thus, it will make the research more powerful. Care was taken to write the tables used in the research in the form of APA6 standard. Writing the results of the research in the form of frequencies, excluding parametric or non-parametric tests, calls into question the reliability of the research. Discussion, conclusion and suggestions section were written in the research. It would be more meaningful to have student opinions and suggestions written accordingly to the research results.
Author Response
Please see the attachment.
- The research seems up-to-date and interesting in terms of its subject. Although the research title is not too long, it is suitable for the purpose. It was also bad that the abstract of this research was not in the structured abstract form. In other words, it would not be good not to write a summary including the purpose, method, data collection tool and results in the summary. However, the research findings should be briefly mentioned in the abstract and the term "we" should not be used. Third language is used in research articles.
Authors’ Response: We thank the reviewer for this comment, and the abstract has been revised accordingly. Our original submission included the purpose and method and it has now been expanded to include the tools used for data collection to be used by students, as well as the associated results of the research. With respect, we feel that the use of first or third person language in scientific research articles is simply down to a matter of preferences rather than convention. Third-person text is certainly not a mandate in the Education Sciences editorial guidelines.However, we have taken the reviewer’s suggestion and changed the abstract to read in the third-person.
- The introduction of the research is appropriate in terms of literature, but it is not sufficient. The introductory part of the research is also not sufficient in terms of subject area. The research is related to STEM, but the concept of STEM has not been adequately addressed. For example, the study "Tezer, M. (2019. The Role of Mathematical Modeling in STEM Integration and Education. In Theorizing STEM education in the 21st century. IntechOpen") in the field of STEM and its theory and similar research could be included. The bibliographies used are up-to-date. Therefore, the use of new bibliography in the introduction and discussion sections of the research has enriched the research.
Authors’ Response: We have now addressed the insufficient background information surrounding STEM education on pages 2 lines 71-77). We also added references concerning this information (28-30 and 46). Additional text and references were also added to the experimental design and overview section (page 4, lines 167-187).
- Purpose and sub-objectives were written in line with the findings. The research method is not well written. Experimental research method was used in the research. But in the research, it would be better to write the experimental research design for this experimental study.
Authors’ Response: The experimental design and overview section has now been amended to include a more detailed explanation of the framework of the study. These additions can be found on page 4 between lines 167-187.
- The research could be used in STEM education, but it could also have an educational aspect. It would be better if the research included students' opinions and the validity-reliability of the data collection tool was also included. In other words, the educational dimension of the research is lacking. My recommendation is to add students' views to the research through qualitative methods. Thus, it will make the research more powerful. Care was taken to write the tables used in the research in the form of APA6 standard. Writing the results of the research in the form of frequencies, excluding parametric or non-parametric tests, calls into question the reliability of the research. Discussion, conclusion and suggestions section were written in the research. It would be more meaningful to have student opinions and suggestions written accordingly to the research results.
Authors’ Response: The goals of this research is the design of a judicious experiment for implementation into a laboratory setting. The inclusion of student opinions would represent the publication of human subject data, which would require a IRB review and approval before implementing the experiment in a classroom setting and obtain students’ opinion and suggestions. The experiment would have to be implemented in future semesters. As such, this would not be achievable within the requisite time frame and would not be possible during the 10 day revision period. Such a study is planned for a future study. As for the table format (see SI), we are reporting the results of our analytical tests (TLC): in the scientific method, multiple trials (at least three) are carried out to obtain a statistically significant mean value and associated uncertainty as required. Repeated trials would allow for minimizing the impact of both systematic and random error, that is, errors arising from limitation of the technique/instrument or operator. Such trials are reported in the tables in the SI.

Reviewer 2 Report
In this work, the authors present a laboratory experiment for undergraduate chemistry students with basic knowledge of general chemistry. It will provide the students with the opportunity of practicing concepts related to solvent extraction, chromatography and spectroscopic analysis and their application in understanding the UV protective molecular filters present in sunscreens. This Reviewer considers that the proposal is highly interesting for teaching purposes and the manuscript is well written. Therefore, I recommend publication of the work. Nevertheless, in order to improve the clarity of some parts, I suggest the authors to address the following points:
- On page 3, at the beginning, revise the grammar of the sentence “In this manuscript, we add the study of the photostability of the prepared of sunscreen samples by means of steady state irradiation with UV excitation light, in tandem with UV/vis spectroscopy as probing technique”.
- At the beginning of Section “Pedagogical Aims and Learning Outcomes”, the authors of this work try to describe the structure of the courses and the academic context of the proposed laboratory experiment. I find this explanation too short and not very general. They should contextualize a bit more at which point of the education of the undergraduate student this experiment should be included. This explanation should not be particular for the undergraduate chemistry curriculum at UL Lafayette but for a general undergraduate chemistry curriculum.
- On page 3, at the end, it is written “(see aim 2)”, but the reader cannot see a list with “Aim 1” and “Aim 2”. I suggest to transform those two paragraphs into a numbered list.
- It would be useful for the reader to remind the definition of SPF by means of a footnote.
- It would increase the quality of the work if the authors briefly summarize the mechanism of photostability (as suggested in the literature) for each of the compounds shown in Fig. 1.
- The use of a light source with a wavelength at 315 nm should be better motivated. Might the use of a higher-energy UV source enhance the differences between the sunscreens?
- Regarding the Supporting Information, the reader might miss the fitting procedure of the absorbance signals.
Round 2
Reviewer 1 Report
Last time, I tried to explain to the authors that this research was far from the educational pillar. Despite the correction made in the Manuscript, no improvement has yet been made in this research regarding its usefulness to students, student attitudes, and conceptual framework in relation to STEM. The new bibliographies used are also far from the educational dimension of STEM. I would like this technical study to be handled together with the educational dimension. Unfortunately, it didn't happen this way. The study should have been reviewed in terms of education, and why it should be done as STEM. Otherwise, you only have STM of STEM.
Author Response
We thank the reviewer for taking the time to evaluate our manuscript carefully and for their suggested additions. We believe that our revised manuscript addressed the educational aspects of this laboratory and its future implementation in an undergraduate laboratory setting. Our manuscript makes it clear that our purpose is to design a robust and novel state-of-the-art experiment aimed at teaching photochemistry in real-time, with the view that interested readers may use the detailed methodology to implement the experiment into their curricula.
The implementation and collection of detailed student satisfaction data, as suggested by this reviewer, will require an additional 1 year, including a lengthy approval process by our internal review board for using human subject data. It is, therefore, impossible to make such a colossal implementation in the 10 days requested by the editorial office. We had already outlined this limitation in the original response letter.